# An Overview of Off-Label Use of Humanized Monoclonal Antibodies in Paediatrics

**DOI:** 10.3390/medicina58050625

**Published:** 2022-04-29

**Authors:** Roberto Bernardini, Gaia Toschi Vespasiani, Arianna Giannetti

**Affiliations:** 1Paediatrics and Neonatology Unit, San Giuseppe Hospital, 50053 Empoli, Italy; 2Specialty School of Paediatrics, Alma Mater Studiorum, University of Bologna, 40138 Bologna, Italy; gaia.toschivespasiani@gmail.com; 3Paediatrics Unit, IRCCS Azienda Ospedaliero-Universitaria di Bologna, 40138 Bologna, Italy; arianna.giannetti@libero.it

**Keywords:** off-label, humanized monoclonal antibodies, paediatrics, children, omalizumab, mepolizumab, dupilumab, reslizumab, benralizumab

## Abstract

In recent years, off-label and unlicensed drug use has extensively developed in the paediatric population. For a long time, clinical trials in the paediatric population were considered complicated to perform because of ethical problems, causing frequent off-label use. Off-label drug use remains an important public health issue, especially for children with rare conditions or with diseases not responsive to conventional treatments. The present paper is a narrative review of the literature of off-label drug use in children. The aim of our study is to summarize the main works dealing with the off-label use of biological drugs in paediatrics. Further studies analyzing their efficacy, safety, and cost–benefit ratios are needed to extend the use of biological therapies to the paediatric population.

## 1. Introduction

The aim of the present review is to evaluate the off-label application of biological therapy in the paediatric population and to update the field of unlicensed drugs in children. In recent years, biologics represent new, promising and more personalized therapeutic options. Over the years, the number of biological therapies approved by the Food and Drug Administration (FDA) for use in the paediatric population has substantially increased. However, the use of many drugs remains off label in children. The term “off label” refers to the use of a drug that is not included in the package insert (approved labeling) for that drug [1]. For a long time, clinical trials in children were considered complicated to perform because of ethical problems, thus, leading to frequent off-label use.

For this purpose, we identified off-label studies for the main drugs used in children.

## 2. Method

### 2.1. Literature Search Methods

We conducted a narrative review of the literature concerning off-label use of monoclonal antibodies. A literature search was performed in October 2021 across MEDLINE/PUBMED to identify studies investigating off-label use of the main monoclonal antibodies. We focused on the use of omalizumab, mepolizumab, dupilumab, reslizumab and benralizumab.

The search criteria were: Omalizumab* OR Mepolizumab* OR Dupilumab* OR Reslizumab* OR Benralizumab* OR Omalizumab* AND children* OR Mepolizumab* AND children* OR Dupilumab* AND children* OR Reslizumab* AND children* OR Benralizumab* AND children* OR Omalizumab* AND off-label* OR Mepolizumab* AND off-label* OR Dupilumab* AND off-label* OR Reslizumab* AND off-label* OR Benralizumab* AND off-label* OR Omalizumab* AND review* OR Mepolizumab* AND review* OR Dupilumab* AND review* OR Reslizumab* AND review* OR Benralizumab* AND review* OR biologic* AND children* OR monoclonal* AND children* OR monoclonal antibodies* AND children* OR monoclonal * AND off-label* OR off-label* AND children* OR off-label drugs* AND children*.

The titles and abstracts of the citations identified from the searches and the content of relevant full texts were evaluated. We focused our research on studies published within the last 5 years. We also searched relevant files from the FDA (www.fda.gov, accessed on 20 October 2021) and EMA (http://www.ema.europa.eu, accessed on 20 October 2021) databases.

### 2.2. Eligibility Criteria

We included studies with the following inclusion criteria: randomized, controlled trials (RCTs); case-control and case-report studies; observational studies in design; target population paediatric patients under 18 years of age; studies published in English.

### 2.3. Exclusion Criteria

We excluded studies with the following criteria: duplicate publications; articles not in full text; articles not written in English; study participants not children.

### 2.4. Data Extraction and Analysis

A narrative approach was chosen to summarize the diverse range of studies in a structured manner. Briefly, studies were first grouped by type of drug and, secondly, by type of disease. For each drug taken into account, we report on doses, dose interval, response to treatment and possible adverse effect.

## 3. Results

### 3.1. Omalizumab

Omalizumab is a recombinant humanized monoclonal antibody that targets the Fc portion of the free IgE antibody and prevents the link of free IgE to high-affinity IgE receptors (FcεRI) on effector cells such as mast cells, basophils and dendritic cells. In this way, it interrupts proinflammatory mediator release, blocks mast cell activation and reduces eosinophil infiltration [2]. In addition, it downregulates FcεRI expression in these cells, which may reduce the allergen presentation to T cells and the Th2-mediated allergic pathway [3].

All of its effects contribute to reducing allergic inflammation and blocking the allergic inflammatory cascade, justifying its clinical efficacy in IgE-mediated diseases. Omalizumab is recommended to be administered as a subcutaneous injection. Omalizumab was first approved in Australia (2002), followed by the United States (2003), the European Union (2005) and Japan (2009), first, for severe, uncontrolled asthma in adults and then in children (12 years old and older) [4]. The dose and frequency of administration in patients with asthma are based on a nomogram that is derived from the serum IgE level and the body mass index (BMI) [5,6].

Since 2016, it has also been approved for patients of 6 years of age and older, with a positive skin test or in vitro reactivity to a perennial aeroallergen and symptoms that are inadequately controlled with inhaled corticosteroids. [5] Both the FDA and the European Medicines Agency approved omalizumab as an add-on therapy for the treatment of chronic spontaneous urticaria (CSU) in adults and adolescents (12 years of age and over) with inadequate response to H1 antihistamine treatment since 2014. Dosing for CSU is not dependent on serum IgE level or body weight [5].

Finally, omalizumab was approved in 2020 for the add-on maintenance treatment of nasal polyps in adult patients of 18 years of age and older with inadequate response to nasal corticosteroids [5]. So, omalizumab in paediatric patients is currently approved only for the treatment of moderate-to-severe, uncontrolled allergic asthma in children aged 6 years and over and chronic spontaneous urticaria (CSU) in adolescents of 12 years of age and older. The use of omalizumab for other IgE-related diseases is, currently, off-label.

Data from clinical trials and case reports/series performed in the paediatric population showed its safety and efficacy in other IgE-related disorders and in children under 12 years. In particular, data from the majority of studies in patient populations, including adults, adolescents and children, showed that omalizumab is well tolerated [7]. We analyze and report the main studies that investigated omalizumab off-label use in IgE-mediated and non-IgE-mediated diseases in Appendix A.

#### 3.1.1. Role of Omalizumab in Urticaria

Chronic urticaria (CU) is defined by the presence of recurrent wheals, angioedema or both, lasting for at least 6 weeks. CSU is a chronic urticarial subtype defined as itchy hives that last more than six weeks, with or without angioedema, without apparent external trigger [8].

The certain pathogenesis of CSU is unknown, but it is assumed to be related to histamine release from cutaneous mast cells and basophils [9]. Before the FDA and EMA approval of omalizumab, non-sedating H1 antihistamines were the only option for patients with CSU. Since 2014, omalizumab has been approved for the treatment of CSU with inadequate response to H1 antihistamines in adults and children 12 years and above with great results. To date, it has not been approved for CSU in children younger than 12 years of age. We collected, in Appendix A, the main works where the off-label use of omalizumab in CSU in children less than 12 years of age was reported. We also report the principal papers which analyze the off-label use of omalizumab in cold and solar urticaria.

#### 3.1.2. Role of Omalizumab in Allergic Bronchopulmonary Aspergillosis

Allergic bronchopulmonary aspergillosis (ABPA) is a severe disease caused by Aspergillus-fumigatus-induced hypersensitivity. This disease may affect patients with asthma and/or cystic fibrosis (incidence 2 to 15%) [10,11]. Actually, the mainstay therapy for ABPA consists of an association of antifungals and systemic CS. Nevertheless, long-term CS treatment can lead to adverse effects, such as diabetes, and, also, many patients do not respond to treatment. This has led to the search for new treatments [12].

In this way, omalizumab is used as a CS-sparing agent to reduce or replace CS use. Until now, just a few case reports (reported in Appendix A) have analyzed the efficacy and safety of omalizumab in ABPA, showing improvement in lung function, CS-sparing effect and a decrease in the frequency of acute exacerbations [13]. Randomized, controlled trials are needed to evaluate further the safety and efficacy of omalizumab in ABPA in the paediatric population.

#### 3.1.3. Role of Omalizumab in Severe Refractory Atopic Dermatitis

Severe refractory atopic dermatitis (AD) is a chronic skin condition associated with elevated serum IgE levels, caused by interaction of genetic, inflammatory and environmental factors. The treatment of AD depends on the severity of the disease.

The first-line therapy is adequate hydration of the skin, in addition to topical CS, topical calcineurin inhibitors and allergen-specific immunotherapy. In more severe cases of AD, treatment includes systemic immunosuppressors (such as cyclosporine, azathioprine, methotrexate and mycophenolate) and systemic CS [14,15]. Antihistamines are used as symptomatic therapy. Due to its safety and efficacy in other IgE-mediated diseases, omalizumab is a promising option for patients with severe and refractory AD. Moreover, it appears to be well tolerated and has good efficacy in patients with severe AD compared to other systemic treatments, leading to a reduction in AD severity and improvement in quality of life, as shown in Appendix A. However, recommendation for its use in clinical practice requires evidence from randomized, controlled trials [16].

#### 3.1.4. Role of Omalizumab in Vernal Keratoconjunctivitis

Vernal keratoconjunctivitis (VKC) is a severe, chronic, ocular disease characterized by corneal conjunctival inflammation with giant conjunctival tarsal papillae and/or limbal inflammation that can result in loss of visual acuity and blindness [17]. Standard treatment includes topical antihistamines, mast cell stabilizers, CS and topical immunosuppressors (cyclosporine or tacrolimus) [18]. Despite these therapies, patients with VKC frequently do not experience improvement in their symptoms. Moreover, due to long-term CS treatment, children often have unacceptable and chronic side effects. Therefore, novel therapies are actually under investigation, including omalizumab.

In Appendix A, we report the main papers on the use of omalizumab in children with VKC, showing, in most cases, improvement in ocular symptoms.

#### 3.1.5. Role of Omalizumab in Autism Spectrum Disorder

Autism spectrum disorder (ASD) is a neurodevelopmental disorder characterized by social disability, communication deficit and repetitive behaviors [19]. Recent literature reported an association between allergic symptoms and ASD severity [20]. ASD patients often have elevated serum IgE levels, which are implicated in the allergic reaction cascade. It is assumed that allergic symptoms may elicit anomalous behaviors through pain and discomfort in children with ASD. Moreover, IgE-mediated mast cell activation and autoimmunity are potential pathogenic factors for ASD [21]. Thus, it has been hypothesized that the pathophysiology of IgE-mediated allergies may exacerbate ASD-related behavioral problems [22]. In this regard, omalizumab, a monoclonal anti-IgE antibody, could attenuate neuropsychiatric symptoms and cause subsequent improvement in cognitive development.

As shown in Appendix A, in 2015, Jyonouchi [23] published a case report which showed that the use of omalizumab improved cognitive development and allergic symptoms. In 2021, Kong et al. [22] confirmed it, showing that allergic symptoms can exacerbate behavioral symptoms, and omalizumab may be a promising symptomatic treatment for children with both ASD and IgE-mediated allergies.

#### 3.1.6. Role of Omalizumab in Chronic Rhinosinusitis with Nasal Polyposis

Chronic rhinosinusitis with nasal polyposis (CRSwNP) is a condition characterized by eosinophilia in the nasal mucosa and peripheral blood and is often associated with asthma. The exact pathophysiology of nasal polyposis is not completely explained, but recent studies demonstrated that it involves cytokines such as IL-5, IL-4 and IL-13. Levels of IgE are often elevated in patients with nasal polyp tissue [24]. In recent years there has been a growing number of papers showing the efficacy of omalizumab for severe nasal inflammation in adults [25,26,27]. Omalizumab has also shown significant improvement in endoscopic, clinical and patient-reported outcomes in severe CRSwNP in patients with inadequately controlled CRSwNP in two replicated phase-3 trials, POLYP 1 and POLYP 2 [28]. In 2019, Shoda et al. analyzed the role of omalizumab in children with CRSwNP, showing no efficacy in reduction of nasal polyposis, despite improving the asthma symptoms [29].

#### 3.1.7. Role of Omalizumab in Food Allergy and Anaphylaxis

A food allergy (FA) is an abnormal immunologic reaction to a food and affects up to 4% of the population [30]. Allergies can be divided into two main categories according to the type of immunological mechanism underlying them: IgE-mediated or non-IgE mediated. IgE-mediated reactions are the most common, presenting with different degrees of severity, ranging from mild oral allergy syndrome to anaphylaxis [30].

Standard treatment of FA was an elimination diet with strict avoidance of the culprit food, leading to an important reduction in quality of life (QoL) for patients with multiple food allergies [31]. In recent years, a novel therapy, known as oral immunotherapy (OIT), has emerged. It is based on the administration of increasing doses of the single food allergen to provide protection against severe, clinical manifestations. In children with multiple food allergies, other therapies should be considered. So, it has been hypothesized that omalizumab, used as a treatment for severe uncontrolled asthma, may increase their tolerance threshold to foods.

Over the years, some trials have been conducted in adults to analyze the effect of omalizumab as a strategy for reducing side effects and speeding up the build-up phase during OIT for peanut [32,33], milk [34,35], egg [36] and multiple food allergies [37,38].

In 2019, Fiocchi et al. showed that omalizumab treatment for asthma increases both the food allergen threshold and QoL of children due to better asthma control and a reduction in dietary restrictions [39]. The cost/benefit ratio of omalizumab for selected cases of FA remains to be evaluated in larger and controlled prospective studies.

#### 3.1.8. Role of Omalizumab in Patients with Asthma and High IgE Levels

Omalizumab has been approved as a treatment for severe uncontrolled asthma in patients of 6 years of age and older. The dose and frequency of its administration are based on a dosing chart that is derived from the serum IgE level and the body mass index (BMI) [5,6]. For children with IgE levels higher than the indicated therapeutic window, only a few studies have examined the drug’s efficacy and safety [40,41,42]. In 2018, Wang et al. showed that omalizumab may also be safely used in children with serum IgE levels above the indicated therapeutic windows [43].

### 3.2. Mepolizumab

Mepolizumab is a humanized IgG1/k monoclonal antibody against interleukin-5 (IL-5), preventing its association with IL-5-receptor (IL-5Rα) on eosinophils [44]. Therefore, through this action, mepolizumab successfully decreases eosinophil numbers in both blood and induced sputum [45]. The FDA [46] approved mepolizumab as an add-on maintenance treatment for patients with severe asthma aged 12 years and older. Since 2019, it has also been approved by the FDA for 6- to 11-year-old children with severe eosinophilic asthma.

Mepolizumab was also approved for the treatment of adult patients with eosinophilic granulomatosis with polyangiitis (EGPA), or Churg–Strauss syndrome by the FDA in 2017 [46] and by EMA. In 2021, EMA recommended granting an extension of indication to mepolizumab as an add-on treatment for patients aged 6 years and older with relapsing–remitting or refractory EGPA [47].

In 2020, the FDA approved mepolizumab as the first and only biologic treatment for adult and paediatric patients aged 12 years and older with hypereosinophilic syndrome (HES) for ≥6 months without an identifiable, non-hematologic secondary cause. In 2021, both the FDA and EMA approved it as an add-on maintenance treatment of adult patients with chronic rhinosinusitis with nasal polyps (CRSwNP). Mepolizumab is recommended to be administered as a subcutaneous injection, and the most common side effects are headache and local reaction at the injection site.

The dose and frequency of administration are the following:Severe asthma in patients aged 12 years and older: 100 mg administered subcutaneously once every 4 weeks;Severe asthma in patients aged 6 to 11 years: 40 mg administered subcutaneously once every 4 weeks;CRSwNP: 100 mg administered subcutaneously once every 4 weeks;EGPA: 300 mg as three separate 100 mg injections administered subcutaneously once every 4 weeks.

We analyze and review the main studies that examined mepolizumab’s off-label use in other diseases in Appendix A.

#### 3.2.1. Role of Mepolizumab in Eosinophilic Esophagitis

Eosinophilic esophagitis (EoE) is a relatively rare condition in which chronic eosinophil inflammation causes symptoms of esophageal dysfunction, such as vomiting, failure to thrive, abdominal or chest pain and dysphagia.

The gold standard for EoE diagnosis is based on biopsy, demonstrating increased intraepithelial esophageal eosinophil (>15 eosinophils per high power field (eos/hpf)) without concomitant eosinophilic infiltration in the stomach or duodenum [48]. The most common treatment for EoE in children includes an elimination diet and proton pump inhibitor (PPI), in combination with topical and systemic CS [48,49].

The role of IL-5 in the pathogenesis of EoE was studied in animal models and, recently, also in clinical reports in adults [50,51,52]. So, in 2011, Assa’ad et al. [53] conducted the first double-blind, randomized, prospective, multicenter clinical trial with mepolizumab in children with EoE younger than 12 years. They demonstrated that IL-5 is involved in the pathogenesis of EoE in children, but a complete response to mepolizumab (defined as <5 eos/hpf at week 12) was only achieved in 8.8% of the studied children. Despite this, mepolizumab reduced esophageal intraepithelial eosinophil counts, confirming the role of IL-5 in the pathogenesis of EoE. Thus, it was hypothesized that additional pathogenetic factors contribute to the disease.

Given that both eosinophils and mast cells play a role in EoE pathogenesis, in 2013, Otani et al. [54] investigated whether mepolizumab reduced esophageal mast cell accumulation in paediatric EoE biopsy specimens from a previous, randomized, anti-IL-5 trial. They showed that 40% of patients responded to mepolizumab (defined as <15 eos/hpf), and 77% of all subjects had decreased numbers of mast cells after the treatment.

#### 3.2.2. Role of Mepolizumab in Hypereosinophilic Syndrome

Hypereosinophilic syndrome (HES) is a rare condition characterized by persistent blood eosinophilia, defined as ≥1500/mm^3^, and eosinophil infiltration into various organs with consecutive tissue damage, most frequently involving skin, lungs, the cardiovascular system and the central nervous system [55]. Most patients respond to CS, which is the first-line therapy. Despite it, CS side effects are frequent, leading to a reduction in QoL and increased morbidity [56]. Immunomodulatory or cytotoxic agents (e.g., IFNa) are recommended in CS-resistant cases and as CS-sparing agents. Given that IL-5 is the most important cytokine for eosinophil maturation, migration, proliferation and survival, humanized anti-IL-5 monoclonal antibodies can be effective in HES. In fact, mepolizumab has proven effective as a CS-sparing agent in adults with HES [56] and in paediatric case reports. In 2020, mepolizumab was approved as the first biologic treatment for both adult and paediatric patients aged 12 years and older with HES for ≥6 months without an identifiable, non-hematologic secondary cause. In Appendix A, we mention the main works concerning the off-label use of mepolizumab in children younger than 6 years.

#### 3.2.3. Role of Mepolizumab in Aspirin-Exacerbated Respiratory Disease

Respiratory disease exacerbated by aspirin and other nonsteroidal anti-inflammatory drugs (NSAIDS) is a disease that is characterized by asthma, chronic rhinosinusitis and nasal polyps, aggravated by the administration of cyclooxygenase enzyme (COX)-1 inhibitors [57]. It is an uncommon condition in children. Its management includes the avoidance of COX-1 inhibitors, along with the use of highly selective COX-2 inhibitors and symptomatic treatment based on the severity of asthma and chronic rhinosinusitis. In case of severe manifestations, biological therapies are required. In Appendix A we mention a case report by Méndez Sánchez et al. [58] in which a child of 12 years benefited from the use of mepolizumab.

#### 3.2.4. Role of Mepolizumab in Refractory Thoracic Conidiobolomycosis

In Appendix A, we also report the first case report which describes the use of mepolizumab for invasive fungal infection in a child with life-threatening thoracic conidiobolomycosis, which is usually poorly responsive to conventional therapies [59]. Yeoh et al. [59] assumed that mepoluzimab, by blocking IL-5, inhibits eosinophil recruitment and degranulation, disrupting the granulomatous eosinophilic inflammation encasing the hyphae of the fungus. Through this action, mepolizumab allows the exposure of the fungus to antifungal drugs.

### 3.3. Dupilumab

Dupilumab is a humanized monoclonal antibody directed against the α subunit of the IL-4 receptor (IL-4Rα) [45,60], thus, preventing both IL-4 and interleukin-13 (IL-13) signaling and, consequently, type 2 inflammation. Dupilumab (Dupixent) was approved by the US FDA [61] in March 2017 for the treatment of adult patients with moderate-to-severe atopic dermatitis (AD) not adequately controlled with other therapies.

In 2019, the marketing authorization for dupilumab in the European Union (EU) was extended from adults to adolescents with moderate-to severe AD who were candidates for systemic therapy. More recently, in 2020, dupilumab was also approved by the FDA [61] for children 6 to 11 years old with moderate-to-severe AD whose disease is not adequately controlled with topical therapies.

Dupixent is also indicated in adults and adolescents 12 years and older as an add-on maintenance treatment for severe asthma with type 2 inflammation characterized by raised blood eosinophils and/or a raised fraction of exhaled nitric oxide (FeNO) which is inadequately controlled with high-dose ICS plus another medicinal product for maintenance treatment. Recently (October 2021), dupilumab was also approved by the FDA [61] for 6- to 11-year-old children with severe asthma.

Dupilumab is also indicated as an add-on therapy with intranasal CS for the treatment of adults with severe CRSwNP for whom therapy with systemic CS and/or surgery does not provide adequate control of the disease.

Dupilumab is the first biologic agent targeting type 2 inflammation. Several studies are being carried out to assess the efficacy and safety of it in adolescents and children with type 2 inflammatory diseases, such as EoE and food allergies. The paediatric dosing regimen is weight-based.

#### 3.3.1. Role of Dupilumab in Alopecia Areata

Alopecia areata (AA) is a common cause of dermatology consultations in children. AA treatment remains challenging, and, often, AA is refractory to conventional therapies. Data from recent studies showed that dupilumab may be an effective treatment for AA when traditional therapies have failed, especially in patients with concurrent AD or asthma, for which the benefits are clear. Some case reports, described in Appendix A, demonstrated improvement with dupilumab therapy. Given its role in inhibiting type 2 inflammation, dupilumab might target AA of the scalp where there can be upregulation of this mechanism [62]. Instead, Chung et al. showed two cases of patients with hair loss while on dupilumab for AD [63]. So, further and larger studies are needed to assess the efficacy and safety of dupilumab for AA.

#### 3.3.2. Role of Dupilumab in Other Skin Diseases

Data from the recent literature showed the role of dupilumab in other skin conditions such as dyshidrotic eczema, eosinophilic annular erythema, prurigo nodularis and actinic prurigo. The exact pathological mechanism underlying these conditions is unknown. It has been hypothesized that there is an activation of the type 2 inflammatory response, providing a suggested mechanism of action for dupilumab in the treatment of eosinophilic dermatoses [64]. The results of these case reports suggest that treatment with dupilumab may be effective in children, but further studies are required.

#### 3.3.3. Role of Dupilumab in EGPA

EGPA, previously known as the Churg–Strauss syndrome, is a chronic disease characterized by asthma, rhinosinusitis, pulmonary infiltrates, neuropathy and eosinophilic vasculitis [65]. CS are currently regarded as the cornerstone for treatment, although most patients are often dependent on systemic CS or other immunosuppressors. Given the side effects associated with long-term therapy with CS and with immunosuppressive agents, there is a need for novel and more efficacious treatment.

First, for adults in 2017, then, in the paediatric population in 2021, mepolizumab was approved as an add-on treatment for patients with relapsing–remitting or refractory EGPA.

Despite mepolizumab showing efficacy in these patients, some others did not reach remission. So, it has been hypothesized that dupilumab can be effective in reducing severe exacerbations in eosinophilic asthma also in patients with EGPA. In 2020, Galant-Swafford et al. reported two successful treatments of antineutrophil cytoplasmic antibody (ANCA)-negative EGPA using dupilumab after progression of disease on mepolizumab [66].

### 3.4. Reslizumab

Reslizumab (Cinqaero; Cinqair) is an IgG4/k humanized monoclonal antibody against IL-5, a cytokine involved in differentiation, maturation and survival of eosinophils and as mediator of eosinophilic airway inflammation [67]. It is indicated as an add-on therapy in patients aged ≥ 18 years with severe eosinophilic asthma inadequately controlled despite high-dose inhaled CS plus another medicinal product for maintenance treatment. In 2016, an intravenous formulation of reslizumab was approved in the USA, Canada and Europe by EMA [68] as an add-on maintenance treatment. Furthermore, reslizumab was included in the most recent iteration of the Global Initiative for Asthma strategy document for the treatment of severe refractory eosinophilic asthma [69].

The efficacy of reslizumab in eosinophilic asthma was evaluated in three randomized, double-blind, placebo-controlled studies (studies I to III) of 16 to 52 weeks’ duration, involving 1268 patients with moderate-to-severe asthma inadequately controlled [70].

EMA [68] reported that 39 paediatric asthma patients between 12 and 17 years were randomized to take reslizumab 0.3 mg/kg, reslizumab 3 mg/kg or placebo as part of two 52-week exacerbation studies (studies I and II) and one 16-week lung function study (study III). No treatment effect on asthma exacerbations was observed for this age group, and, given the small sample size and baseline imbalances resulting from a subgroup analysis, no conclusion could be drawn regarding asthma efficacy in the paediatric population. Thus, reslizumab is not yet approved in the paediatric population.

A recent review of the literature by Busse et al. evidenced that all anti-IL-5-pathway-specific therapies are more effective than a placebo in reducing exacerbations and improving symptom control in children with asthma [71] (Appendix A).

#### Role of Reslizumab in Eosinophilic Esophagitis

Although reslizumab is not yet approved for the treatment of EoE in children, evidence from the multicenter, randomized trial showed improvement in histologic disease activity. There are no data concerning the long-term safety and efficacy of this treatment in children and adolescents with EoE.

In 2011, Spergel et al. [72] evaluated the effect of Reslizumab in children and adolescents with EoE, with long-standing disease. They found out that patients who received Reslizumab showed statistically significant and clinically relevant reductions in esophageal eosinophil count compared with those who received placebo, but Reslizumab was not associated with significant differences in assessments of clinical symptoms and quality of life [73]. Reslizumab appears to be safe in children with eosinophilic esophagitis.

### 3.5. Benralizumab

Benralizumab is an anti-interleukin-5 receptor alpha (IL-5Rα) monoclonal antibody which was approved by FDA in 2017 [74] for severe eosinophilic asthma in patients 12 years and older and by EMA for adults. While the other anti-IL-5 biologics (mepolizumab and reslizumab) directly bind to IL-5, blocking the activation of the eosinophils, benralizumab has the added mechanism of inducing eosinophils and basophils apoptosis in tissues through antibody-dependent, cell-mediated cytotoxicity (ADCC) [75].

Benralizumab has to be administered subcutaneously at a dose of 30 mg every 4 weeks for 3 doses, followed by every 8 weeks thereafter. It is, overall, well tolerated with its described safety profile.

In Appendix A, we report off-label use of Benralizumab in children with pulmonary eosinophilia, which represents a group of heterogeneous disorders consisting of lung disease and eosinophilia in the peripheral blood, alveoli and/or pulmonary interstitium [76].

We also mention the first case of a paediatric patient with idiopathic HES associated with significant end-organ damage successfully treated with Benralizumab [77].

## 4. Discussion

This narrative review was conducted to explore the literature regarding off-label use of humanized monoclonal antibodies in children. To our knowledge, this is the first literature narrative review to evaluate off-label use of the main humanized monoclonal antibodies in the paediatric population.

Off-label drug use remains an important public health issue, especially for children with rare diseases. Evidence, not label indication, remains the gold standard for clinicians when making therapeutic decisions for their patients. In fact, we decided to focus our research on the main humanized monoclonal antibodies used in clinical practice, drawing from both case reports/series and reviews/RCT. We designed a table for each drug (Appendix A), enumerating the main studies reported in the literature to facilitate clinicians who have to take decisions for their patients.

Taking into account each drug and each study, we reported in the tables the doses, dose interval, response to treatment and possible adverse effects, leading to a clear and wide view of the monoclonal antibodies landscape.

Most of the drugs prescribed have not been tested in children, and their safety and efficacy are often supported by a low quality of evidence. Therefore, a large percentage of prescriptions for children in clinical, daily practice is used off label. New and better-designed studies are needed to evaluate long-term safety and efficacy of monoclonal antibodies in children. Further, RCTs should be conducted to better analyze potential benefits and assess possible adverse effects of the use of these drugs.

Novel ways should be achieved by competent authorities to promote more research in this field.

## 5. Conclusions

From this review, it is evident that off-label drug use is very common in the paediatric population. Most biological drugs are still off label for children because of ethical problems in conducting clinical trials in the paediatric population, leading to an important public health issue. In Table 1, we summarize the label and off-label uses of the main humanized monoclonal antibodies used among children.

Further studies including clinical efficacy, safety and cost–benefit ratios as primary outcomes are needed to extend the use of biological therapies to the paediatric population. Particularly, RCTs, systematic review and meta-analysis could help clinicians when making therapeutic decisions for their patients.

Off-label prescribing is indispensable for many diseases not responsive to conventional treatments, and, because of this, there is still a lot of work to do to ensure the best therapeutic decision making in paediatrics.

## Figures and Tables

**Table 1 medicina-58-00625-t001:** Summary box.

Drugs	Approval FDA Use in Children	Off-Label Use (Including Case Reports)
Omalizumab	Chronic idiopathic urticaria in adults and adolescents (12 years of age and over) with inadequate response to H1 antihistamine treatment Moderate-to-severe uncontrolled allergic asthma in children aged 6 years and over	Chronic idiopathic urticaria in children under 12 years Solar urticaria ASD and allergy disease Allergic bronchopulmonary aspergillosis in cystic fibrosis patients Food allergies High immunoglobulin E levels in patients with asthma Severe atopic dermatitis Chronic rhinosinusitis and nasal polyposis Vernal keratoconjunctivitis
Mepolizumab	Add-on maintenance treatment for patients with severe asthma aged 6 years and older Relapsing–remitting or refractory EGPA for patients aged 6 years and older HES for patients aged 12 years and older	Eosinophilic esophagitis HES in children under 12 years Aspirin-exacerbated respiratory disease Refractory thoracic conidiobolomycosis
Dupilumab	Moderate-to-severe AD not adequately controlled with topical therapies for patients aged 6 years and older Severe asthma for patients aged 6 years and older	Alopecia areata Dyshidrotic eczema Eosinophilic annular erythema Prurigo nodularis Actinic prurigo ANCA-negative EGPA
Reslizumab	Severe eosinophilic asthma inadequately controlled for patients aged 18 years and older	Eosinophilic esophagitis
Mepolizumab	Severe eosinophilic asthma in patients 12 years and older	Pulmonary eosinophilia

## Data Availability

The study did not report any data.

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
