# Peer review of "An Overview of Off-Label Use of Humanized Monoclonal Antibodies in Paediatrics"

_medicina, 2022, doi:10.3390/medicina58050625_

Round 1

Reviewer 1 Report

  1. The title does not represent the contents of the manuscript. It should be changed to "An overview of the off-label use of humanized monoclonal antibodies in pediatric practice"
  2. The numbering of subtitles is confusing and not suitable, and modification is highly suggested.

Author Response

  1. The title does not represent the contents of the manuscript. It should be changed to "An overview of the off-label use of humanized monoclonal antibodies in pediatric practice". We change it
  2. The numbering of subtitles is confusing and not suitable, and modification is highly suggested. Done

The aim of this review study is to summarize clinical studies evaluating the off-label use of biological drugs in paediatrics. It is an interesting subject and the review would provide helpful information about biological use in real settings in children. There were 5 drugs in this review : omalizumab, Mepolizumab, Dupilumab, reslizumab, benralizumab. We change the title. We focus on these drugs because are the most common used in pediatrics. We also decide to do this because of the large number of papers/reviews/RCTs/case series regarding humanized monoclonal antibodies.

Reviewer 2 Report

The aim of this review study is to summarize clinical studies evaluating the off-label use of biological drugs in paediatrics. It is an interesting subject and the review would provide helpful information about biological use in real settings in children. There were 5 drugs in this review : omalizumab, Mepolizumab, Dupilumab, reslizumab, benralizumab.

I have some major comments

  1. The review should be better organized in standard sections as introduction, method, result discussion and conclusion

For example:

  1. Introduction
  2. Method
  • Results
    1. Omalizumab
    2. Mepolizumab
    3. Dupilumab
    4. Reslizumab
    5. Benralizumab
  1. Discussion:

The discussion could be made for each drug as in the result

The strength and limitation of the review must be discussed

  1. Conclusion
  2. It would be more rigorous to do a systematic review and follow the PRISMA guideline for each biological drugs. These 5 drugs should be stated clearly in the objective.
  3. For my opinion, it would not be relevant to include case report. But if the authors want to keep all of studies, it would be appreciated to summary all case report in 1 table and in supplementary file.
  4. In the method section : even though it is a narrative and non-exhaustive review, for the transparency, strategies of search and the flowchart of search results must be provided at least in supplementary files. Criteria to include studies must be described….Was there any language restriction, double check for screening, selection, extraction of data?
  5. A table to summary all indications approved for each drugs in children would facilitate the lecture and reduce the text

Author Response

I have some major comments

  1. The review should be better organized in standard sections as introduction, method, result discussion and conclusion. Done

For example:

  1. Introduction
  2. Method
  • Results
    1. Omalizumab
    2. Mepolizumab
    3. Dupilumab
    4. Reslizumab
    5. Benralizumab
  1. Discussion:

The discussion could be made for each drug as in the result.

The strength and limitation of the review must be discussed

  1. Conclusion
  2. It would be more rigorous to do a systematic review and follow the PRISMA guideline for each biological drugs. These 5 drugs should be stated clearly in the objective. Regaring such as broad topic, we decided to do a narrative review, focusing our research on the most used humanized monoclonal antibodies. Actually, in the method section, we have specified our choice.
  3. For my opinion, it would not be relevant to include case report. But if the authors want to keep all of studies, it would be appreciated to summary all case report in 1 table and in supplementary file. Done
  4. In the method section : even though it is a narrative and non-exhaustive review, for the transparency, strategies of search and the flowchart of search results must be provided at least in supplementary files. Criteria to include studies must be described….Was there any language restriction, double check for screening, selection, extraction of data? Done
  5. A table to summary all indications approved for each drugs in children would facilitate the lecture and reduce the text: Done